# Atrial Natriuretic Peptides as a Bridge between Atrial Fibrillation, Heart Failure, and Amyloidosis of the Atria

**DOI:** 10.3390/ijms24076470

**Published:** 2023-03-30

**Authors:** Farzad Rahbar Kouibaran, Mario Sabatino, Chiara Barozzi, Igor Diemberger

**Affiliations:** 1Department of Medical and Surgical Sciences, University of Bologna, 40138 Bologna, Italy; farzad.rahbarkoui2@unibo.it (F.R.K.); chiara.barozzi@unibo.it (C.B.); 2Unit of Heart Failure and Transplantation, IRCCS Azienda Ospedaliero-Universitaria di Bologna, 40138 Bologna, Italy; mario.sabatino2@gmail.com; 3Unit of Cardiology, IRCCS Azienda Ospedaliero-Universitaria di Bologna, 40138 Bologna, Italy

**Keywords:** atrial natriuretic peptides, heart failure, atrial fibrillation, cardioversion, ablation therapy

## Abstract

ANP is mainly synthesized by the atria, and upon excretion, it serves two primary purposes: vasodilation and increasing the renal excretion of sodium and water. The understanding of ANP’s role in cardiac systems has improved considerably in recent decades. This review focuses on several studies demonstrating the importance of analyzing the regulations between the endocrine and mechanical function of the heart and emphasizes the effect of ANP, as the primary hormone of the atria, on atrial fibrillation (AF) and related diseases. The review first discusses the available data on the diagnostic and therapeutic applications of ANP and then explains effect of ANP on heart failure (HF) and atrial fibrillation (AF) and vice versa, where tracking ANP levels could lead to understanding the pathophysiological mechanisms operating in these diseases. Second, it focuses on conventional treatments for AF, such as cardioversion and catheter ablation, and their effects on cardiac endocrine and mechanical function. Finally, it provides a point of view about the delayed recovery of cardiac mechanical and endocrine function after cardioversion, which can contribute to the occurrence of acute heart failure, and the potential impact of restoration of the sinus rhythm by extensive ablation or surgery in losing ANP-producing sites. Overall, ANP plays a key role in heart failure through its effects on vasodilation and natriuresis, leading to a decrease in the activity of the renin-angiotensin-aldosterone system, but it is crucial to understand the intimate role of ANP in HF and AF to improve their diagnosis and personalizing the patients’ treatment.

## 1. Introduction

In recent decades, the understanding of the role of natriuretic peptides (NPs) in cardiac pathophysiology has improved considerably, increasing their importance both for diagnostic and therapeutic purposes. The natriuretic peptide family consists of three peptides: atrial natriuretic peptide (ANP), brain natriuretic peptide (BNP), and C-type natriuretic peptide (CNP), and they represent the main cardiac endocrine function, as far as we currently know. Among these peptides, ANP is mainly synthesized by the atria, where the precursor form of these peptides split into prohormones and is stored in secretory granules in atrial cardiomyocytes [1,2]. Upon excretion, they are further fragmented into NT-ProANP and ANP. Secretion of ANP occurs due to stretching of the atrial walls and serves two primary purposes: vasodilation and increasing renal excretion of sodium and water [3]. Among patients with heart disease, there is a significant correlation between the mechanical and endocrine function of the heart, which may serve as an indicator of the disease’s stage and severity. Because natriuretic peptides play an essential role in cardiac physiology and pathophysiology, a deep understanding of the regulation of these biomarkers in a cardiac system under normal and abnormal conditions could pave the way for more accurate diagnosis and improved therapies, particularly in heart failure (HF) and atrial fibrillation (AF) which are strictly interconnected in a sort of chicken egg dilemma [4]. The role of ANP as one of the products of endocrine function should be carefully studied to understand the regulation of this peptide concerning mechanical function. In patients with AF, the most frequent sustained tachyarrhythmia, the irregular contractions of the left atrium led to stretching of the atrial wall and consequent release of ANP.

## 2. Diagnostic Applications of ANP

The diagnostic value of natriuretic peptides in cardiovascular disease, including AF and HF, has been extensively studied. ANP excretion is primarily regulated by mechanical stretch of the atrium during increased exercise, as observed in heart failure, and may also be influenced by the rate of atrial contraction. ANP release can also be stimulated by factors such as myocardial ischemia, adrenergic agonists, angiotensin II, and vasopressin. The half-life of ANP in plasma is approximately 2–4 min, and it is inactivated by neutral endopeptidase and lysosomal degradation. Elevated ANP levels are associated with left ventricular diastolic dysfunction and left ventricular end-diastolic pressure and may be a risk factor for AF recurrence. ANP levels also correlate with left atrial volume (LA) as determined by echocardiography and cardiac magnetic resonance. After an AF episode, plasma ANP levels decrease rapidly in response to filling pressure, and a steady normalization of ANP concentration occurs in association with improvement in atrial mechanical function [5].

Measurement of NP levels can be useful in the clinical management of cardiovascular disease. In patients with HF, NP is a tool for diagnosis, prognosis, and therapy management, and there are established cut-off values. In addition, NP is useful in the management of AF by screening for new-onset AF, predicting the success of cardioversions and pulmonary vein isolations, and assessing the risk of stroke [6]. In a study by Shiroto et al. [7], they investigated the value of ANP for predicting AF in ischemic stroke patients and they concluded that ANP could be used as a predictor of paroxysmal atrial fibrillation.

ANP levels are typically very low in healthy individuals (20 pg/mL), but increase significantly in HF or AF patients. Measurement of the bioactive form of ANP can be challenging due to its short half-life (2 min), whereas the more stable N-terminal prohormone form (NT-proANP) is readily degraded. Therefore, the mid-region of NT-proANP (MR-proANP), which is less susceptible to degradation, is used in the clinical evaluation of HF patients. Measurement of the bioactive form of ANP can be challenging due to its short half-life (2 min) and the more stable N-terminal prohormone form (NT-proANP) is readily degraded. Therefore, the mid-region of NT-proANP (MR-proANP), which is less susceptible to degradation, is used in the clinical evaluation of HF patients [8]. Figure 1 [6], shows different NPs with their characterization and more importantly the half-life which is a crucial parameter in clinical diagnosis.

In a study, Rossi et al. [9] aimed to determine the relationship between AF and natriuretic peptide activation and found that patients with AF had similar symptoms, comorbid conditions, cardioactive medications, pulmonary pressure, left atrial volume, and left ventricular (LV) ejection fraction and filling characteristics compared to those in sinus rhythm but higher NT-proANP levels (2613 ± 1681 vs. 1654 ± 1323 pg/mL, *p* = 0.007). The higher NT-proANP levels persisted even after adjustment for underlying heart disease (*p* < 0.0001). In another study, Büttner et al. [10], investigated the association between NT-proBNP and NT-proANP levels with three phenotypes of AF progression: persistent AF, left atrial diameter dilatation (LAD), and left atrial low-voltage areas (LVA). The results showed that NT-proANP levels were significantly higher in LVA patients and correlated with LAD, whereas NT-proBNP levels were not significantly associated with any of the phenotypes. NT-proANP levels also increased according to the four disease progression groups: paroxysmal AF without LVA, persistent AF without LVA, paroxysmal AF with LVA, and persistent AF with LVA. This study suggests that natriuretic peptides have differential sensitivity to AF progression phenotypes, and the clinical implications of NT-proANP should be investigated in larger studies. In a recent study, Büttner et al. [11], Measured NT-proANP levels in AF patients and compared them with non-AF individuals and they found that peripheral and cardiac NT-proANP levels were significantly higher in AF patients than in non AF controls. In multivariable analysis, NT-proANP remained significantly different between non-AF individuals and AF patients. This study also showed that NT-proANP levels were higher in cardiac blood samples than in peripheral blood from AF patients and that the ability to predict LVAs using both cardiac and peripheral NT-proANP levels was modest.

There are also many studies on the most stable fragment of pro-ANP, namely MR-proANP in HF or AF patients, showing the diagnostic value of this peptide [12]. Gohar et al. [13] evaluated the diagnostic value of MR-proANP in a community cohort of men and women with newly suspected heart failure (HF) and reported that the addition of MR-proANP to a previously validated clinical model improved the c-statistic from 0.82 to 0.86, and with the addition of NT-proBNP, the c-statistic was 0.87, whereas the best exclusion cut-point for MR-proANP was 40 pmol/L, with a sensitivity of 0.99, specificity of 0.06, PPV of 0.30, and NPV of 0.92. On the other hand, Latini et al. [14] have investigated the prognostic value of MR-ProANP and they showed that this biomarker could be used as a predictor of AF recurrence in patients with sinus rhythm and AF history. Although it is difficult to predict the recurrence of atrial fibrillation (AF) in patients with a history of AF, this study found that the higher the concentration of NT-proBNP, the earlier the onset of AF, and changes over time in the concentration of MR-proANP were associated with subsequent recurrence of AF.

The diagnostic value of MR-proANP was also reported by Yagmur et al. [15] in critically ill patients with or without sepsis. They measured MR-proANP plasma levels of patients upon admission to the medical intensive care unit (ICU) and compared the results with healthy controls. They reported that MR-proANP levels were significantly higher in ICU patients with sepsis and correlated with inflammatory cytokines, markers of organ dysfunction, and several adipocytokines. High MR-proANP levels were also associated with mortality risk in ICU patients. The study concluded that MR-proANP is a potential biomarker for organ dysfunction, sepsis, and mortality risk in critically ill patients.

Last but not least, Schweizer et al. [16] in a recent study measured MR-proANP levels in 1759 patients with acute ischemic stroke and demonstrated the association between MR-proANP levels and stroke. Interestingly, they were able to derive and validate a model to predict newly diagnosed AF (NDAF) based on age and MR-proANP levels.

In conclusion, ANPs have important diagnostic applications in AF and HF. They have been shown to be useful biomarkers for the diagnosis and severity assessment of these diseases. ANPs may also be a useful tool for predicting the risk of hospitalization and mortality in patients with HF. However, further studies are needed to explore the clinical utility of ANPs in these conditions.

## 3. Therapeutic Applications of Natriuretic Peptides

Since the PARADIGM randomized control trial showed that inhibition of neprilysin (NEP) improved outcomes in patients with HF, understanding the role of natriuretic peptides in this context has gained increasing interest [17]. Data from this study showed that during treatment with sacubitril/valsartan, BNP levels remained unchanged, whereas NT-proBNP levels and urinary cyclic GMP (cGMP) increased. These observations were explained by the effect of inhibition of NEP on the activity of its substrate BNP and its second effector cGMP. Conversely, the decrease in NT-proBNP levels was associated with the effects of the drugs on cardiac loading conditions and wall stress [18].

On the basis of these assumptions, several studies have examined the effect of sacubitril/valsartan on natriuretic peptide activity. Data from the PROVE-HF and EVALUATE HF studies confirmed a sustained decrease in NT-proBNP associated with positive remodeling as measured by left ventricular ejection fraction and left atrial dimensions [19,20].

On the other hand, the association between inhibition of NEP and BNP levels was weaker, and the change in the effector pathway of cyclic guanylate cyclase was somewhat independent of BNP trajectory [21]. Two studies showed relatively low reproducibility of available diagnostic assays for BNP as a potential explanation for the former observation [21,22]. Furthermore, Nougué et al. [23], pointed out that BNP is a poor substrate for NEP and higher levels could even inhibit its activity. In their study of 73 patients treated with sacubitril/valsartan, they showed that the drug did not affect BNP levels or BNP-derived cGMP signaling. They hypothesized that the observed decrease in NT-proBNP was related to increased glycosylation of proBNP and subsequent different handling of this substrate rather than to increased BNP levels. Of note, the observation that the clinical effect of NEP inhibition is not fully explained by an increase in BNP levels or activity suggests a role for the other natriuretic peptides. From this point of view, some studies provided interesting mechanistic insights focusing on the potential role of ANP in this scenario.

For example, the aforementioned study by Nougué et al. confirmed a strong inhibition of the activity of NEP and showed a sustained 4-fold increase in ANP levels during sacubitril/valsartan therapy. Similarly, a biomarker substudy [24] of the PROVE-HF trial, which included 111 patients with longitudinal analysis of ANP levels confirmed an early and sustained increase in ANP after initiation of sacubitril/valsartan therapy. The increase was detectable as early as 14 days and persisted up to 45 days. Furthermore, the increase in ANP level was followed by an increase in urinary cGMP, and a greater increase in ANP level was associated with improved cardiac remodeling, with a slight increase in left ventricular ejection fraction and a reduction in left atrial volume. A smaller study somewhat confirmed these findings, highlighting that after switching from sacubitril/valsartan to valsartan, there was a decrease in ANP and an increase in NT-proBNP can be observed, whereas BNP levels remained largely unchanged [25].

Overall, these findings support the concept that ANP has an underlying biological effect in heart failure and those strategies that restore its physiological properties could improve the clinical outcomes. This concept applies to NEP inhibition, because ANP appears to be a more potent substrate for NEP and a more potent activator of the cGMP pathway than BNP but could as well pave the way for more targeted ANP substitutive therapy. Such strategies are currently under investigation and could in the future open a new chapter in the treatment of heart failure [26].

## 4. The Role of ANP in Heart Failure

Heart failure (HF) is a clinical and pathophysiological syndrome caused by ventricular dysfunction, leading to a different combination of volume overload and decreased cardiac output. The hemodynamic changes of HF lead to activation of renin-angiotensin-aldosterone and the sympathetic nervous system to restore homeostasis. Activation of neurohormonal systems causes sodium and water retention along with vasoconstriction to restore blood flow. However, in chronic heart failure, these mechanisms lead to a dysfunctional adaptation that further exacerbates hemodynamic derangement and pathological cardiac remodeling.

Natriuretic peptides play an important role in heart failure through their effects on vasodilation and natriuresis, leading to a decrease in the activity of the renin-angiotensin-aldosterone system [27]. Among NPs, ANP is secreted mainly by the atria, whereas BNP is secreted by the ventricles in response to a ventricle overload. Because different conditions such as atrial fibrillation or left ventricular dysfunction have different effects on overload and wall tension, it is important to understand the different secretion patterns. Previous studies have shown that only ANP—but not BNP—levels are associated with left atrial pressure in isolated atrial overload (i.e., mitral stenosis), whereas both NPs are associated with left ventricular filling pressure in chronic heart failure [28]. In addition, Rossi et al. [9] showed that AF is the most important determinant of ANP levels independent of structural heart disease, whereas BNP levels are strongly associated with the degree of left ventricular dysfunction, but not with AF. Therefore, despite common physiological pathways, analysis of ANP and BNP levels may have different purposes and meanings in diseases affecting the atria, ventricles, or both. In one study [29], a radioimmunoassay was performed to analyze the relationship between ANP and cardiac filling pressure in four groups of normal patients, in patients with cardiovascular disease and normal filling pressure, in patients with elevated filling pressure without HF, and patients with elevated filling pressure and HF. The results showed that HF had a direct effect on the elevation of ANP levels as well as cardiac filling pressure, and they noted that “congestive heart failure reflects not an ANP deficiency state, but rather a compensatory increase in peptide release”. BNP has a well-established diagnostic and prognostic role in heart failure, whereas ANP is recognized as a prognostic marker for the success of ablation of atrial fibrillation, especially in the absence of structural heart disease. Because of their antagonistic effect on neurohormonal activation in heart failure, natriuretic peptides may also have a therapeutic effect. In particular, research has focused on restoring their physiological properties, which appear to be disrupted in heart failure. Indeed, the development of sacubitril, a neprilysin inhibitor that prevents BNP degradation, opened a new therapeutic avenue in heart failure. Of note, the addition of sacubitril to valsartan showed in the PARADIGM randomized controlled trial a significant reduction in mortality and HF hospitalization among patients with heart failure with reduced left ventricular ejection fraction [17]. Therefore, the association of sacubitril and valsartan is now a cornerstone in the treatment of this condition. The potential therapeutic role of ANP is less defined, as its physiological effects on heart failure are more nuanced and studied to a lesser extent.

In heart failure, volume overload leads to atrial distension, which promotes the release of ANP in response to these hemodynamic changes. However, it has been reported that patients with heart failure may have a significant deficiency of ANP and an attenuated renal response to ANP, leading to retention of sodium and water [5]. According to a review by Abassi et al. [3], what is considered ANP deficiency is not the case because the highest concentrations of circulating ANP occurs in congestive heart failure (CHF), and disease progression further increases plasma ANP levels. In another review, Charloux et al. [30] addressed the problem of decreased natriuresis despite high ANP levels in chronic heart failure by explaining renal resistance to ANP. As shown in Figure 2 [30], the progression of chronic heart failure from a compensated to a decompensated state results in increased ANP levels but, in contrast, decreased natriuresis has been observed.

Renal ANP resistance is caused by several mechanisms, including decreased ANP availability at the renal site, decreased binding of ANP to its receptor (NPR), or degradation of ANP at the renal site; these mechanisms can result in sodium and water retention at elevated ANP levels. Interestingly, both left ventricular dysfunction and renal failure, which is promoted by impaired renal response, are important markers of heart failure, to the extent that renal failure is considered one of the strongest markers of mortality [30]. Several complex mechanisms are involved in the regulation of ANP in the cardiac and renal systems. It is therefore crucial to improve our understanding of ANP in the regulation of neurohormonal hemostasis in HF and AF. Indeed, the progression of left ventricular dysfunction could lead to mechanical failure of the left atrium, resulting in the development of atrial fibrillation. Conversely, cardiac fibrosis, particularly in the atria in patients with long-standing AF, may also lead to a reduction in ANP synthesis and natriuresis. Therefore, the interplay between ANP and atrial fibrillation is extremely complex and not yet fully understood. From a therapeutic point of view, a small study [31] recently investigated the role of the exogenous ANP analogue carperitide in patients with acute heart failure. The diuretic effect of carperitide correlated inversely with ANP levels and was more pronounced in patients with preserved left ventricular ejection fraction, who in turn had lower ANP baseline. These preliminary findings support the role of ANP as a novel therapeutic target in heart failure, particularly in the setting of ANP deficiency or resistance.

## 5. The Role of ANP in Atrial Fibrillation

AF is the most common cardiac arrhythmia worldwide with multifactorial etiology. Several factors contribute to the disease, including aging, left atrial enlargement, left atrial dysfunction, heart failure, respiratory conditions, or mitral valve disease. AF induces irregular contractions in the atrium, resulting in electrical and structural remodeling. AF is also reported to be associated with endothelial damage, inflammation, and fibrosis [5,32,33]. Since left atrial dysfunction is directly related to AF, the focus should be on ANP as an indicator of the degree of atrial stretch. In the following sections, we will discuss the relationship between AF and ANP.

Atrial fibrillation causes the atria to lose efficiency in pumping blood to the ventricles (i.e., atrial systole), resulting in increased stress on the atrial walls and mechanical stretch. Therefore, ANP is secreted to stabilize hemodynamics through natriuresis and vasodilation, and this increase is directly related to the increase in left atrial stretching. Although ANP concentration increases during AF, it has been reported that a prolonged AF leads to a decrease in ANP level, due to the structural changes in the left atrium leading to a loss of ANP production sites [5]. Maarten et al. [34] addressed this issue by analyzing ANP levels during long-standing AF and reported that a prolonged AF contributes to atrial remodeling, which leads to structural damage and may decrease the ability of the atria to produce ANP. This phenomenon can be seen in Figure 3 [34].

As shown in Figure 3, the ANP concentration decreases with the duration of AF. The hypothesis is that during this time, there may be a phenomenon of fibrosis with a decrease in ANP-producing cells, resulting in a lower ANP concentration. In addition, a cardiac surgery study [35] demonstrated a relationship between ANP and fibrosis, indicating that the amount of ANP is inversely related to the extent of fibrosis in the atria. ANP may prevent fibrosis, but if fibrosis occurs in the meantime due to multiple recurrences of AF, ANP concentration continues to decrease. Eventually, this process leads to atrial remodeling and loss of mechanical and electrical function. Of note, a study by Cao et al. [36] analyzed gene expression, ANP protein, and production of fibrosis-related proteins in the right atrial appendages (RAAS) of patients with paroxysmal and persistent AF and sinus rhythm. They showed that NPs protein gene expression was increased in AF patients but to a lesser extent in persistent AF. On the other hand, the tissue’s NPs protein storage and plasma levels were comparable between AF subgroups, whereas the fibrosis-related protein expression was higher in persistent AF. These results suggest that ANP gene expression is reduced in advanced atrial remodeling (i.e., persistent AF) in parallel with an increase in fibrosis-related markers, but that protein storage and secretion are preserved, at least initially. Conversely, a study of patients undergoing catheter ablation showed that NT-proANP levels were higher in patients with longstanding AF compared with those with paroxysmal AF [10].

These seemingly contradictory results should be considered in light of the different patient population, as catheter ablation is generally performed in subjects with less advanced atrial remodeling, and the difficulty in defining the natural history of atrial remodeling in AF. Detailed assessment of the time course of loss of atrial “integrity” will be an important milestone in this topic, and further studies are needed. Interestingly, ANP production itself could contribute to the development of atrial fibrosis by amyloid deposition. Amyloid deposits (misfolded proteins) can result from aging and various diseases. Among the types of amyloidosis, isolated atrial amyloidosis (IAA) is characterized by misfolded ANP proteins. Maarten et al. [37] discussed the effects of amyloidosis on AF and vice versa and suggested a “vicious circle” between atrial amyloidosis and AF where ANP amyloid deposits are more common in patients with AF than with sinus rhythm. However, the most critical question is the exact relationship between IAA and AF [37]. Since ANP in IAA tends to aggregate and accumulate in the atria, these misfolded proteins could increase fibrosis and lead to AF. On the other hand, higher ANP levels in AF patients could increase the likelihood of ANP misfolding, leading to amyloidosis. These concepts were further explored in studies that investigated both the role of ANP oligomers on the electrophysiological properties of atrial cells and the effects of mutant ANP. Indeed, specific mutations in the ANP gene are known to be associated with increased susceptibility to AF and known forms [38]. Accordingly, Yang et al. [39] showed that ANP oligomers exert detrimental effects on the electrical properties of atrial cells and that this phenomenon is enhanced in the presence of mutant ANP. These findings are corroborated by Hua and colleagues [40], who confirmed an increased susceptibility to the development of AF in an animal model with mutant ANP.

However, despite this potential mechanism, strong clinical evidence is lacking, and the interplay between these entities and their pathophysiology is yet to be elucidated. These complex interactions are illustrated in Figure 4 and are worth further investigation and understanding.

### 5.1. Recovery of the Endocrine and Mechanical Function after Cardioversion

Current treatments for AF focus on restoring the sinus rhythm of the heart, which is not as effective because atrial remodeling and structural damage are the leading causes of AF, which can be irreversible if not diagnosed and treated in time [41,42]. An important tool for the diagnosis and prognosis of AF is the study of patients’ endocrine and mechanical functions before and after treatment. In a study of 42 patients with chronic AF undergoing cardioversion, Wozakowska-Kaplon et al. [43] found that the average ANP concentration in AF patients was reduced by approximately two-fold within 24 h after successful cardioversion (to the same level as in the control group with sinus rhythm), whereas ANP levels remained the same in the group with unsuccessful cardioversion (Figure 5).

In another study, Nishino et al. [44] analyzed the recovery time of hormonal function and mechanical and electrical function after cardioversion. They found that hormonal function, assessed by plasma ANP concentration, recovered within 24 h after cardioversion, whereas peak atrial velocity in transmitral flow (an indicator of mechanical function) recovered between 1 and 4 weeks. Another study by Thomas et al. [45] confirmed these results by measuring ANP level and peak A-wave velocity before and after DC cardioversion and showed delayed recovery of mechanical function compared with recovery of ANP level. The results are shown in Figure 6.

According to these results, the recovery of atrial contraction (mechanical function) takes about one month. In contrast, the concentration of ANP (as an indicator of endocrine function) drops rapidly and recovers within a week. With a similar behavior, peak A-wave velocity should increase to higher values as ANP drops to almost zero within 4 h. However, we have no such increase in mechanical function, showing a mismatch between recovery times. The point of view here is that due to the rapid drop in ANP concentration (within 4 h) in comparison with left atrial stretch and mechanical dysfunction, acute heart failure may occur.

### 5.2. Effects of Ablation Therapy on ANP

There are numerous studies on the effects of catheter ablation on ANP and its prognostic role. Sacher et al. [46] studied ANP levels in persistent/permanent AF patients undergoing ablation therapy and reported that ANP decreased significantly the day after restoration of sinus rhythm. In contrast, however, recovery of mechanical function was found to take more than three months. In another study [47], cardiac mechanical and endocrine function was assessed four months after radiofrequency ablation therapy, and there was improvement in maximum left atrial volume and ANP values. However, no improvement in left atrial ejection fraction was observed.

An interesting finding that emerged from the studies is the kinetics of ANP after catheter ablation. These data [48] showed that there was a transient increase in ANP levels shortly after the procedure that gradually decreased thereafter. These observations suggest that direct damage to atrial cells releases more ANP. An important unanswered question is whether there is a preferential location of ANP-producing cells and whether ablation in a selected zone could somehow preserve atrial endocrine function. However, the clinical relevance of such a hypothesis is not known. A study by Nakanishi et al. [49] suggests that in patients with AF, undergoing catheter ablation, increased ANP levels are associated with favorable atrial remodeling. Similarly, a study of epicardial AF ablation [48] showed that in patients with long-standing persistent AF, increased ANP levels were associated with persistence of the sinus rhythm despite a low success rate. Conversely, a meta-analysis [50] showed that higher ANP levels were an independent predictor of AF recurrence, but only in patients without structural heart disease. These conflicting observations highlight the complexity of this issue and reflect the heterogeneity of patients with AF, underpinning the concept of a more personalized approach to risk stratification.

Data from both cardioversion and catheter ablation consistently demonstrated delayed recovery of mechanical and endocrine atrial function estimated from ANP kinetics. These findings provide the basis for formulating hypotheses about the interplay between these aspects of atrial physiology after restoration of the sinus rhythm. Data from surgical treatment of atrial fibrillation with the Maze procedure [51] show that pulmonary congestion can occur after the procedure, and although the loss of mechanical function due to surgical trauma plays an important role, some authors suggest that rapidly declining ANP levels also contribute. Acute pulmonary congestion occurs in 3–4% of patients treated with electrical cardioversion [52]. Given the aforementioned differences between endocrine and mechanical atrial function, this phenomenon could at least play a role in acute heart failure following cardioversion. Although data supporting the mechanistic effect are lacking, further studies on these subjects could further improve our understanding of atrial physiology and even pave the way for more accurate patient management in this clinical scenario.

## 6. Conclusions

The production of natriuretic peptides represents the major endocrine function of the heart. While the role of BNP in the diagnosis and treatment of HF is currently well established, the assessment of ANP levels is still a subject of research without any clinical implication. Among the various factors hampering this, such as a shorter half-life, the most important is most likely the higher complexity of the endocrine function of the atria combined with several unresolved questions in the available literature regarding the mechanism of regulation during different diseases (such as HF and AF), the presence of a specific site for ANP secretion, the specific interplay between filling pressure, atrial contraction, and the presence of fibrosis. The intriguing link between AF, ANP, and atrial amyloidosis can be a possible explanation not only for the AF phenomenon [53], but we may hypothesize that it may also play a role in patients with HF with preserved ventricular function, especially in case of untreated hypertension. Loss of endocrine function may explain the lack of benefit of comprehensive AF ablation beyond pulmonary vein isolation and the occurrence of acute AF after electrical cardioversion or maze surgical procedure. Despite the long time that has passed since its identification, it appears that ANP has many more “stories” to tell if we have the time and patience to piece together its complex puzzle.

## Figures and Tables

**Figure 1 ijms-24-06470-f001:**
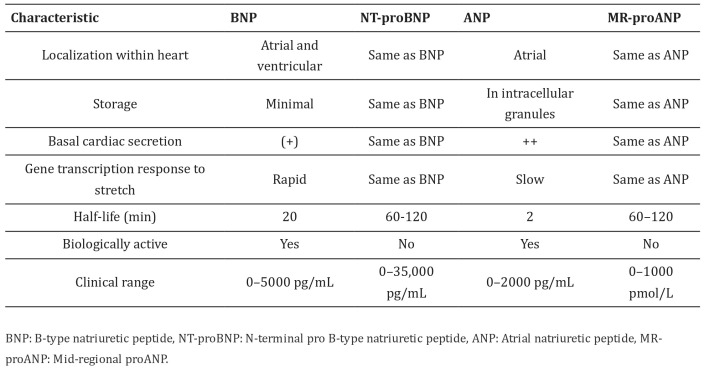
Summary of NPs’ characteristics [6]. The symbols “(+)” and “++” make a comparison between the secretion of each peptide.

**Figure 2 ijms-24-06470-f002:**
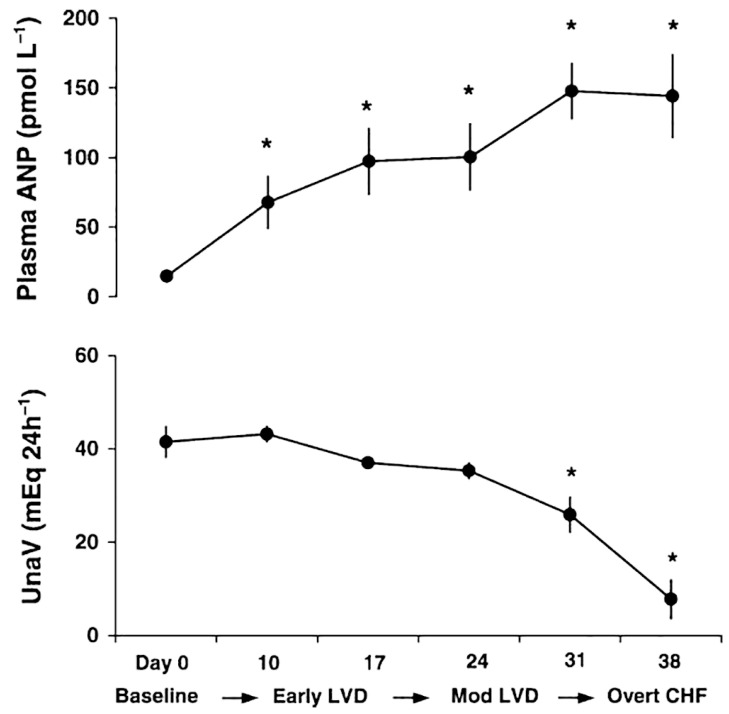
Tempora changes in circulating ANP and urinary sodium excretion (UNaV) in left ventricular dysfunction (LVD) progression [30]. * *p* < 0.05 compared with the control group.

**Figure 3 ijms-24-06470-f003:**
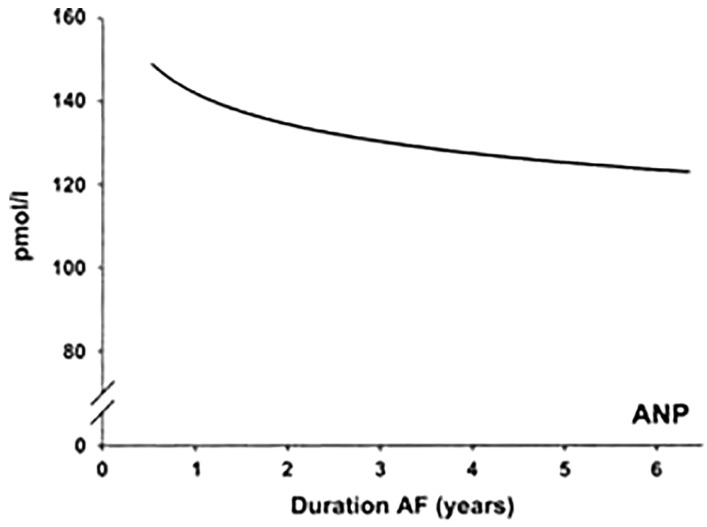
Correlations between the duration of atrial fibrillation (AF) and plasma concentrations of ANP [34].

**Figure 4 ijms-24-06470-f004:**
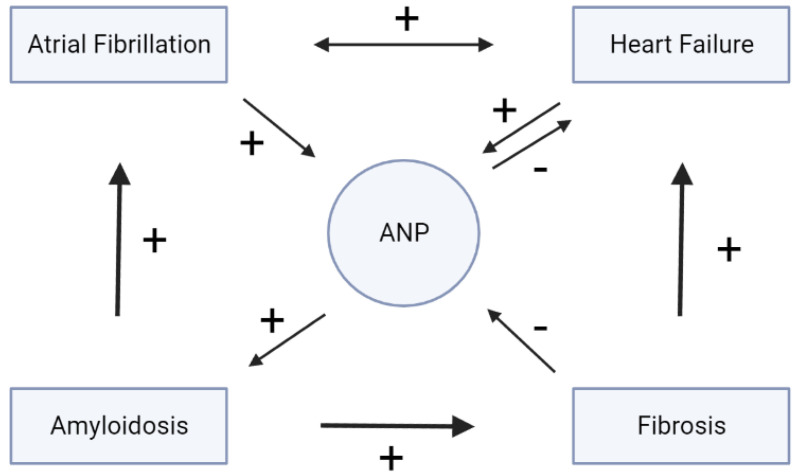
Relationship among AF, HF, and amyloidosis; Arrows with a (+) sign show an increase/promote effect, and arrows with a (−) sign show a reduction effect.

**Figure 5 ijms-24-06470-f005:**
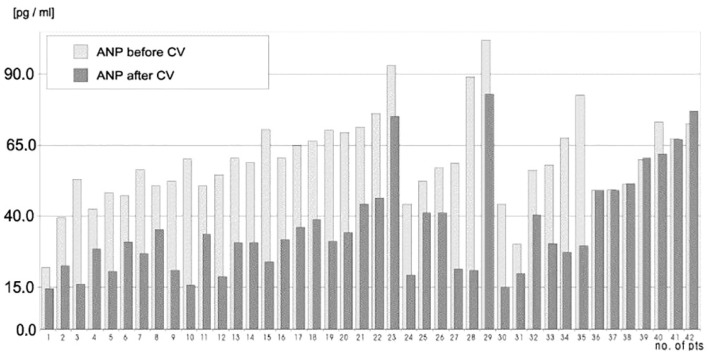
ANP levels in patients with AF before and after cardioversion (cardioversion in patients 36–42 was unsuccessful) [43].

**Figure 6 ijms-24-06470-f006:**
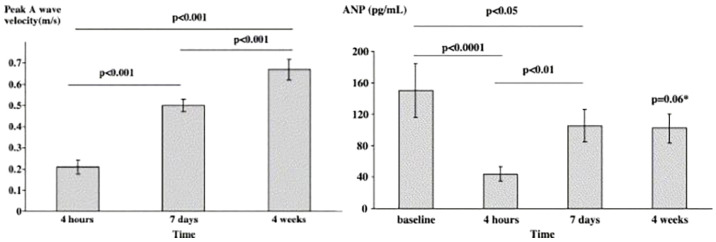
Mean plasma levels of ANP (**Right**) and mean peak A wave velocity (**Left**) before and after cardioversion, * *p*-value versus baseline [45].

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
