# Peer review of "Atrial Natriuretic Peptides as a Bridge between Atrial Fibrillation, Heart Failure, and Amyloidosis of the Atria"

_ijms, 2023, doi:10.3390/ijms24076470_

Round 1
Reviewer 1 Report
A state-of-the-art paper summarising the current role of atrial natriuretic peptides in cardiology.
I would like to congratulate the Authors the comprehensive presentation of the material. The manuscript is well written, references are appropriately selected.
I would like to raise 2 issues of potential clinical implications:
- The Authors state that 'ANP is recognized as a prognostic marker for the success of ablation of atrial fibrillation, especially in the absence of structural heart disease' (line 190-192). Do any studies identify ANP concentration thresholds that would predict success of ablation? Similarly, are there any thresholds that would predict the success of electric cardioversion?
- A growing interest in identification of short-lasting episodes of AF/supraventricular arrhythmia is being observed. Do the Authors know any studies linking the presence of such episodes with ANP concentration?
Author Response
We appreciate the time and the valuable points by the reviewer and we uploaded the response to all points as a pdf file in attachments.

Reviewer 2 Report
Dear authors,
I have a few questions and suggestions:
1. Do you have approval of the authors and journals of figures 1, 2 and 3 for reuse of their material. For figure 2 which is a modified version of another figure in your reference you have to ask the primary authors as well.
2. I find it a bit disappointing to use reviews to write a review. I would have preferred that you study the original articles and write your review based on them. You have done this partially but not completely.
3. What do you mean by resinualization in the abstract?
4. In diagnostic applications of ANP in the third paragraph, second sentence is confusing. "However, measurement of the bioactive form of ANP can be challenging due to its short half-life (2 minutes), whereas the more stable N-terminal prohormone form (NT -proANP) is readily degraded." Using however and then whereas and then saying the N form is readily degraded does not make sense.
5. In role of ANP in heart failure in the 3rd paragraph last sentences you say: "As shown in Figure 2 [27], the progression of chronic heart failure from a compensated to a decompensated state results in increased ANP levels but, in contrast, decreased natriuresis that is not due to ANP deficiency." You already say ANP is increased, so remove the last part of the sentence "that is not due to ANP deficiency".
6. Figure 4 is confusing, and the text describing it is also not that clarifying. The arrows should represent the effects: increase, decrease?mutation?
7. Point one applies for figures 5 and 6 as well.
Best regards.
Author Response

(The authors gave the same response as above.)

Round 2
Reviewer 2 Report
Dear authors,
Hi,
Thanks for implementing the suggestions.
Best regards.